# Unusual Case of Pancreatic Adenocarcinoma with Bladder Metastasis

**DOI:** 10.3390/medicina56120708

**Published:** 2020-12-18

**Authors:** Giorgia Arcovito, Iosè Di Stefano, Laura Boldrini, Francesca Manassero, Jacopo Durante, Alessio Tognarelli, Pinuccia Faviana

**Affiliations:** 1Department of Surgical, Medical, Molecular Pathology and Critical Area, University of Pisa, 56126 Pisa, Italy; giorgia.arcovito@gmail.com (G.A.); Iose@outlook.it (I.D.S.); laura.boldrini@med.unipi.it (L.B.); 2Department of Translational Research and New Technologies in Medicine and Surgery, University of Pisa, 56126 Pisa, Italy; f.manassero@ao-pisa.toscana.it (F.M.); jacopodurante@live.it (J.D.); alessio.tognarelli@gmail.com (A.T.)

**Keywords:** bladder metastasis, immunohistochemistry, metastatic pancreatic

## Abstract

*Background*: The pancreas can be the site of neoplasms of several histogenetic origins; in most cases, tumors derive from the exocrine component, and ductal adenocarcinoma certainly prevails over the others. This tumor displays remarkably aggressive behavior, and it is often diagnosed at a late stage of disease. *Case presentation*: We discuss the rare case of a 76-year-old male with locally advanced pancreatic head adenocarcinoma who developed uncommon metastatic disease. The bladder constitutes a very rare site of metastases, mostly deriving from melanoma, gastric, lung and breast cancers. The bladder’s secondary involvement in pancreatic malignancies represents an extremely unusual occurrence, and there are very few cases described in the literature to date. *Conclusions:* The finding of pancreatic adenocarcinoma metastases leads to a poor prognosis, and patients who are diagnosed at this stage constitute 53% of cases, with a 5-year survival of 3%. Although rare, therefore, the diagnostic hypothesis of pancreatic ductal adenocarcinoma (DAC) metastases to the bladder must, in some cases, be considered, especially if accompanied by a clinical picture that may suggest it.

## 1. Introduction

Ductal adenocarcinoma (DAC) is the most common neoplastic disease of the pancreas [1], accounting for 90% of all pancreatic malignancies [2]. It constitutes 3% of all malignant neoplasms and represents the ninth most frequent cancer in women and the tenth in men [3]. The prevalent site of onset is represented by the head of the pancreas (60–70%), followed by the body-tail (20–25%) and widespread involvement of the organ (10–20%) [4]; for this reason, the presence of the tumor is often associated with dilation of the choledochus duct [5] and/or the Wirsung duct, giving the typical radiological image of the “double duct sign” [6]. The first symptoms are due to the mass effect and the consequent obstruction of the biliary tract, causing abdominal pain, nausea, weight loss, jaundice, itching, dark urine and acholic stools [7]. Diagnosis is often made at advanced stage of disease, since the symptoms are typically scant at the beginning [8]. This occurrence, along with the intrinsic biological features, justifies the dramatically high mortality of this type of cancer [9], ranking seventh among the causes of cancer death worldwide [10]. When the carcinoma is still operable, the five-year survival is 37%, while it dramatically collapses in the metastatic phase (3%). However, the percentage of people receiving early-stage diagnosis is extremely low, standing at 10% [3]. The operability cut-off corresponds to the T3N1M0 stage, which indicates a pancreatic mass with a diameter >4 cm but without involvement of the large pancreatic vessels [11,12]. The histology of pancreatic adenocarcinoma is marked by considerable aggressiveness, which leads to an early local invasion and rapid systemic spread [13]; the most common site of metastasis is represented by the liver, affected in 40–50% of cases [14]; other organs frequently involved are the lungs, bones, adrenals and, contiguously, the stomach, duodenum, transverse colon and left kidney [15,16]. Among the target organs of pancreatic DAC, the bladder certainly represents an exceptionally rare site [15] and it is involved in advanced stage, often in conjunction with the condition of peritoneal carcinosis. Bladder metastases may be the result of hematogenous or “dripping” dissemination [16]. In the sporadic cases of metastases to the bladder described in the literature so far, the most frequent symptom was found to be hematuria. Pancreatic DAC metastases to the bladder are therefore very infrequent, and they can mimic primary urothelial malignancies. In this case report, we will describe the case of a patient with pancreatic DAC diagnosed in advanced stage of disease, who eventually developed bladder metastases and hematuria. This case is one of very few cases described in the literature to date [17].

## 2. Case Presentation

Our clinical case concerns a 76-year-old male with locally advanced pancreatic head adenocarcinoma. In anamnesis, systemic arterial hypertension under treatment, insulin-treated diabetes mellitus II, chronic obstructive pulmonary disease, acute myocardial infarction and stage III chronic kidney disease are found. The oncological history reports a diagnosis of bladder papilloma, treated with transurethral resection of a bladder tumor (TURBT) and subsequent courses of intravesical chemotherapy. The diagnosis of pancreatic cancer was made in 2019 following admission to the U.O. of General Surgery and Transplantation of the AOUP of Pisa. The patient was subjected to observation because of general malaise associated with worsening jaundice. On this occasion, a thoraco-abdominal CT scan was performed, which revealed the presence of an expansive lesion of the head of the pancreas, located in close proximity to the superior mesenteric artery and infiltrating the superior mesenteric vein (Figure 1).

During hospitalization, an external–internal biliary drainage (DBEI) was placed for palliative purposes. An echo-endoscopy was also performed in order to characterize the lesion highlighted on CT, but the procedure was not conclusive; the pathological diagnosis highlighted the presence of non-cohesive epitheliomorphic cellular elements with nuclear atypia. Immunohistochemical investigations showed a positivity for Cytokeratin 7 (CK 7). The patient was discharged after 27 days of hospitalization with DBEI drainage and placed within an appropriate oncological follow-up; moreover, he was directed to chemotherapeutic treatment according to guidelines. Eight months after discharge, the patient visited the emergency department of the same polyclinic due to the occurrence of hematuria. On physical examination, he was apyretic and his hemodynamic parameters were stable. A complete abdominal ultrasound was also performed which highlighted a 5 mm wall thickening of the right floor of the bladder with associated modest right pyelic dilation. The patient was then transferred to the geriatrics ward of the same polyclinic, where, following the replacement of the biliary drainage, he developed a clinical–laboratoristic picture compatible with sepsis, likely to depart from the biliary tract, successfully treated with extensive intravenous broad-spectrum antibiotic therapy. Due to persistent macrohematuria and consequent anemia, the patient was transferred to the O.U. of Urology and underwent blood transfusion and TURBT of the bladder lesion highlighted on ultrasound. A bladder biopsy was thus performed at the urology department of the same polyclinic. Two sections stained with hematoxylin–eosin were obtained from the biopsy fragments. On microscopic observation, the presence of clearly malignant cells, with moderate architectural differentiation, was observed in the context of the suburothelial connective tissue (Figure 2A–C). The lesion infiltrated the bladder wall to full thickness in both sections and was made up of elements with marked cytological atypia, mostly organized in acinar-like glandular structures. These structures showed an irregular morphology, with distorted lumens and sketches of ramifications. Inside the lumens, cellular debris were seen, along with neutrophilic granulocytes. Focuses of small irregular glands and single pleomorphic cells immersed in the stroma were also noted peripherally, as if from tumor budding. The stroma showed a desmoplastic appearance, with a marked amount of fibroblasts and collagen and scattered lymphocytes. The cells showed an irregular, roughly cuboidal, morphology and possessed an eosinophilic cytoplasm. The nuclei appeared increased in size, with irregular contours, marked pleomorphism and hyperchromasia; nucleoli were sometimes prominent. Even with due caution, based on these morphological findings, we were more oriented towards the diagnosis of an adenocarcinoma. To identify the origin of the lesion, immunohistochemical investigations were then carried out which showed a positivity for CK7, Carbohydrate antigen 19.9 (CA19.9) and CDX2 (Figure 3A–C). The hypothesis of an intestinal and prostatic adenocarcinoma was ruled out by the negativity of Cytokeratin 20 (CK20) and PSA, respectively (Figure 3D,E). The immunophenotype was also negative for GATA3, which was useful for excluding a primitive urothelial neoplasm with reasonable certainty (Figure 3F). The immunohistochemical profile appeared suggestive for a bilio-pancreatic primitivity of the lesion, and the case was eventually diagnosed with extrinsic infiltration of the bladder wall by ductal adenocarcinoma of the pancreas. The anatomic–pathological diagnosis was fully compatible with the already known clinical–radiological picture and was therefore confirmed.

## 3. Discussion

Our article deals with the extremely rare case of bladder metastasis from pancreatic ductal adenocarcinoma. If the lesion appeared with a modest degree of differentiation and with a glandular cyto-architecture, these morphological findings suggested a differential diagnosis with adenocarcinomas most frequently responsible for secondary bladder involvement, as well as the variants of urothelial carcinoma. The latter hypothesis was the least convincing from the beginning, since the lesion of the wall had no continuity with the urothelium, which was structurally intact and morphologically normal. Furthermore, the histology of the lesion seemed to suggest a glandular histogenesis, the cells being organized in clear, albeit not regular, ductal structures. In light of the diagnostic difficulty, accentuated by the fact that the evaluation was carried out on biopsy, it was necessary to set up a broad-spectrum immunohistochemical investigation, able to make a differential diagnosis with the main metastatic neoplasms to the bladder, as well as with the rare but plausible case of a variant of urothelial carcinoma or a primary adenocarcinoma. The first evaluation was performed on CK7, CK20 and CDX2, to exclude the statistically more probable hypothesis of intestinal primitivity. The results showed a positivity for CDX2 and CK7, while the CK20 was found to be negative. This last finding, together with the positivity for CK7, allowed us to exclude not only an intestinal origin of the lesion but also, with reasonable certainty, urothelial. In fact, as already highlighted, a peculiar characteristic of urothelial carcinoma is the relatively frequent co-expression of CK7 and CK20 [18]. Furthermore, if it has been ascertained that urothelial carcinomas with glandular differentiation express CK20 in a smaller percentage of cases [19], the expression of CDX2 is not typical of these neoplasms. Moreover, although, as already observed, CDX2 may be found in the intestinal subtype of bladder adenocarcinoma [20], in the examined case, the morphology of the lesion did not show similarities with the intestinal one, which led us to exclude this rare occurrence. On the other hand, CDX2 is most often expressed in colon adenocarcinoma, but this expression should normally be accompanied by that of CK20, in the absence of CK7 [21]. Therefore, excluding the first two hypotheses, the immunohistochemical diagnostic investigation further proceeded, testing PSA, GATA3 and CA19.9. The choice of this panel was aimed, respectively, to exclude the hypothesis of an extrinsic infiltration of a prostate adenocarcinoma, to strengthen the conclusion of a non-urothelial origin of the lesion and to take into consideration the rare but concrete possibility of a secondary coming from the bilio-pancreatic district. The immunohistochemical profile of pancreatic adenocarcinoma is characterized by the positivity for CKs normally expressed by the cells of the ductal epithelium of the pancreas: CK7, 8, 18 and 19 [11]. Regarding CK20, great variability is observed in the expression profiles, and, in most cases, it is not expressed [22]. The glycoproteins CA19.9 and CEA, on the other hand, are frequently expressed by pancreatic malignant cells [23]. Considering the low suspicion based on the morphological characteristics of the lesion, the negativity of PSA and GATA3 justified the exclusion of the hypothesis of prostatic primitivity and urothelial carcinoma. The expression of CA19.9, in light of the results of the IHC previously performed and the reasonable clinical suspicion, provided an immunohistochemical profile that was fully compatible with that of a pancreatic adenocarcinoma. There are very few cases of pancreatic DAC metastases to the bladder in the literature [15,17]. Mostly, patients were female and already harbored peritoneal carcinosis. The favored site of metastases was the posterior wall of the bladder, and the most frequent symptom was hematuria [17]. Although rare, therefore, the diagnostic hypothesis of pancreatic DAC metastases to the bladder must, in some cases, be taken into account, especially if accompanied by a clinical picture that may suggest it.

## Figures and Tables

**Figure 1 medicina-56-00708-f001:**
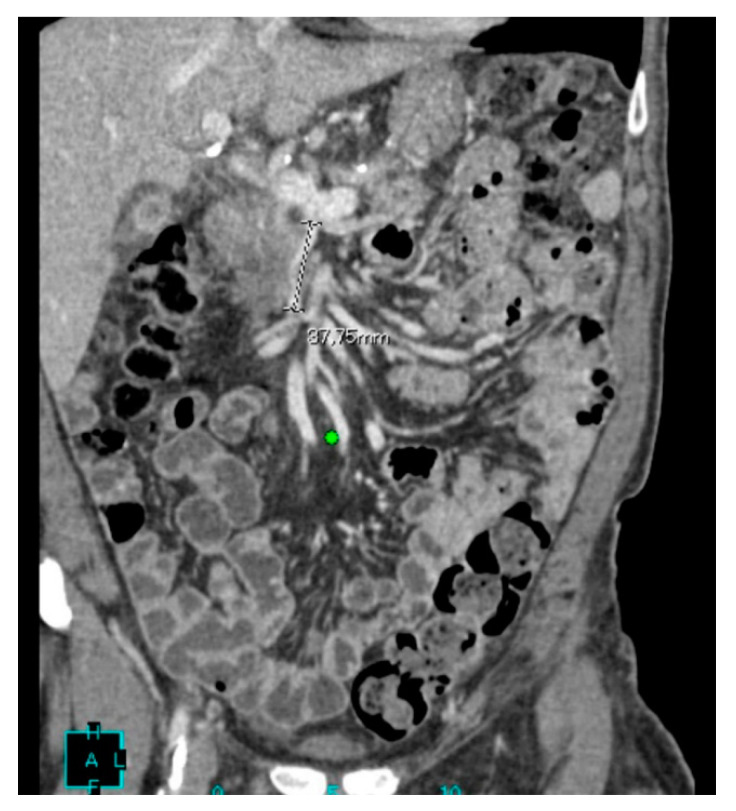
Frontal CT shows an expansive lesion of the pancreas head with irregular margins and indissociable from the superior mesenteric vein crushed for 38 cm.

**Figure 2 medicina-56-00708-f002:**
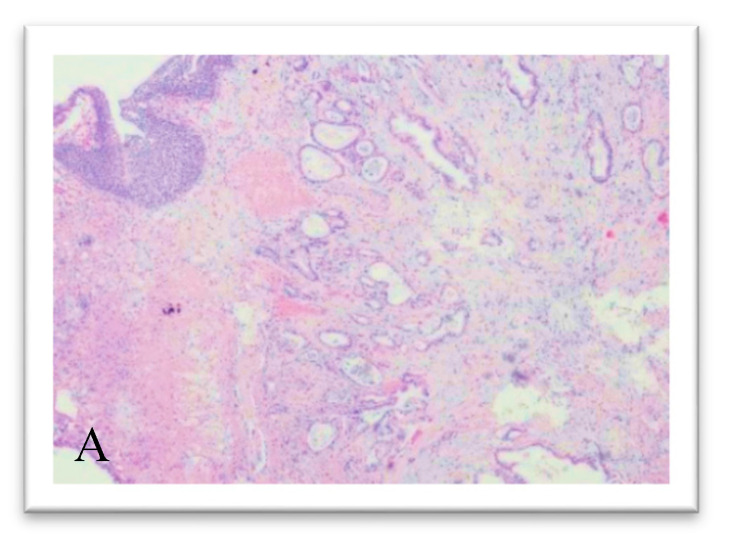
Hematoxylin and eosin, original magnification 4× (**A**) and at magnification 20× (**B**,**C**): the lesion infiltrates the bladder wall to full thickness and is made up of elements with marked cytological atypia, mostly organized in acinar-like glandular structures.

**Figure 3 medicina-56-00708-f003:**
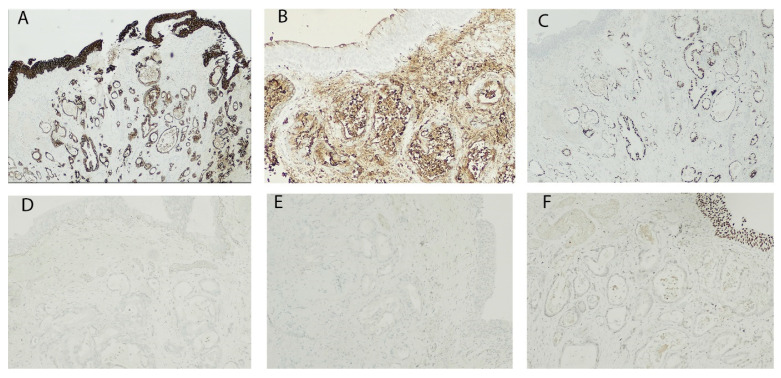
Immunohistochemistry shows positivity for Cytokeratin 7 (CK7) (**A**), Carbohydrate antigen 19.9 (CA19.9) (**B**) and CDX2 (**C**); negativity for Cytokeratin 20 (CK20) (**D**), PSA (**E**) and GATA-3 (**F**); original magnification: (**A**,**C**) 4×, (**B**,**D**,**F**) 20×, (**E**) 10×.

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
