# Peer review of "Unusual Case of Pancreatic Adenocarcinoma with Bladder Metastasis"

_medicina, 2020, doi:10.3390/medicina56120708_

Round 1

Reviewer 1 Report

only a few comments:

  • Do the authors specify the origin of pancreatic cancer: endocrine or exocrine?
  • Pancreatic and duodenal homeobox 1 is a marekr that could help...:

    Generation of Functional Beta-Like Cells from Human Exocrine Pancreas

    If you could present a better quilty of images than what is presented in the manuscript

Author Response

Dear Professor

The tumor mentioned in the case report is an adenocarcinoma that originates and develops in the exocrine pancreas, for which Pancreatic and duodenal homeobox 1 cannot help. We also took new photos to highlight the infiltration of the bladder wall by pancreatic adenocarcinoma.
Thanks

Reviewer 2 Report

This is an interesting case report of pancreatic adenocarcinoma with bladder metastasis. This is an unusual case and this report have an educational impact.

Minor point:

The discussion part is too long. Please focus on the points the authors want to emphasize and shorten it.

Author Response

Dear Professor
Thanks for the suggestion we have, in fact, reduced the discussion of the manuscript, focusing on the case analyzed by us.
Thanks
Best regards